# AID-Purifier: A Light Auxiliary Network for Boosting Adversarial Defense

Duhun Hwang [* 1]   Eunjung Lee [* 1]   Wonjong Rhee [1 2]

## Abstract

We propose an *AID-purifier* that can boost the robustness of adversarially-trained networks by purifying their inputs. AID-purifier is an auxiliary network that works as an add-on to an already trained main classifier. To keep it computationally light, it is trained as a discriminator with a binary cross-entropy loss. To obtain additionally useful information from the adversarial examples, the architecture design is closely related to information maximization principles where two layers of the main classification network are piped to the auxiliary network. To assist the iterative optimization procedure of purification, the auxiliary network is trained with AVmixup. AID-purifier can be used together with other purifiers such as PixelDefend for an extra enhancement. The overall results indicate that the best performing adversarially-trained networks can be enhanced by the best performing purification networks, where AID-purifier is a competitive candidate that is light and robust.

## 1. Introduction

Deep neural networks are vulnerable to adversarial examples generated by adding imperceptible adversarial perturbations to the original examples (Szegedy et al., 2013). To address this problem, various adversarial defense schemes have been proposed, where a vast majority of them can be grouped into three categories. The first category is *gradient masking* (Xiao et al., 2020; Athalye et al., 2018). The second category is *adversarial training* (Madry et al., 2017; Zhang et al., 2019; Lee et al., 2020). The third category is *adversarial purification* (Samangouei et al., 2018; Song et al., 2018; Meng & Chen, 2017; Shi et al., 2021).

In this study, we focus on the third category, adversarial purification. Our objective is to develop a computationally

light and easily attachable purifier such that it can be utilized as an add-on. Specifically, we show that we can boost the performance of Madry et al. (2017); Zhang et al. (2019), and Lee et al. (2020) with a light auxiliary network named *AID-Purifier*. AID-Purifier utilizes **AV**mixup, **I**nformation maximization principles, and **D**iscriminative task as the underlying foundations. Before describing AID-Purifier, we first summarize previous works on adversarial purification methods.

Adversarial purification modifies input examples to increase adversarial robustness, and four well-known purification methods are shown in Figure 1. In (a), a *denoising purifier*, MagNet (Meng & Chen, 2017), is shown. It uses an auto-encoder, called a reformer, as an auxiliary network. The resulting network as a whole, however, becomes just another feedforward network that is vulnerable to auxiliary-aware white-box attacks (Tramer et al., 2020). In (b), a *generative purifier*, Defense-GAN (Samangouei et al., 2018), is shown. Defense-GAN is not easy to train, and its performance is worse than that of another well-known generative purifier. In (c), another generative purifier, PixelDefend (Song et al., 2018), is shown. PixelDefend is computationally heavy owing to its pixel-wise operation. In (d), a *self-supervised-learning-based purifier*, SOAP (Shi et al., 2021), is shown. SOAP yields competitive robust accuracy against state-of-the-art adversarial training and purification methods, but it needs to be jointly trained with the main classifier $C$. Of the four purification methods in Figure 1, SOAP is the only one that requires joint training and thus cannot be used as an add-on.

We herein propose a *discriminative purifier* named AID-Purifier. To the best of our knowledge, this is the first successful purification method based on a discrimination task. AID-Purifier uses an auxiliary discriminator network $D$ to project $x_{adv}$ to a purified example $x_{pur}$ that belongs to a low $p_{adv}(x)$ region. Compared to the four methods in Figure 1, AID-Purifier is distinct because it has all the advantages of the four methods. Unlike denoising purifiers, it is robust against auxiliary-aware attacks. Unlike generative purifiers, it requires light computation and is easy to train. Unlike SOAP, it is an add-on that can be attached to any frozen state-of-the-art network. AID-Purifier is an effective stand-alone defense method; however, it can also create synergies with adversarially-trained networks or other

---

*Equal contribution [1]Department of Intelligence and Information, Seoul National University, Seoul, South Korea [2]AI Institute, Seoul National University, Seoul, South Korea. Correspondence to: Wonjong Rhee <wrhee@snu.ac.kr>.

*Accepted by the ICML 2021 workshop on A Blessing in Disguise: The Prospects and Perils of Adversarial Machine Learning.* Copyright 2021 by the author(s).

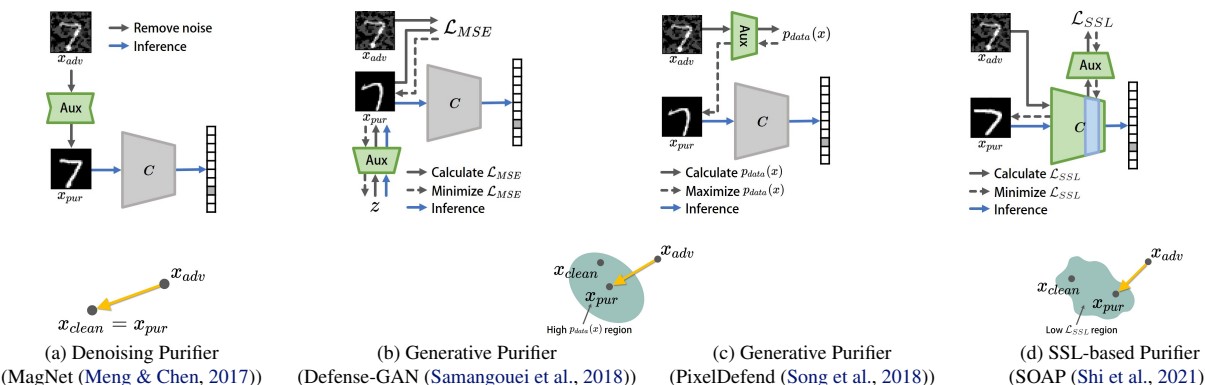

*Figure 1.* Summary of four existing purifiers. The upper diagrams show algorithm overviews. We denote main classification network as $C$, auxiliary network as Aux, network with frozen weights in gray, and network to be trained in green. The lower diagrams show the conceptual relationships between $x_{clean}$, $x_{adv}$, and $x_{pur}$.

purifier networks such as PixelDefend. For all experiments we performed, AID-Purifier was able to boost the robustness of state-of-the-art adversarial training and purification methods.

## 2. Related Works

### 2.1. Detecting adversarial examples with an auxiliary network

For humans, it is difficult to tell the difference between a clean example and its adversarial example. The difference, however, can be detected well by training a binary classification network (Gong et al., 2017; Metzen et al., 2017). A standard binary cross-entropy (BCE) loss can be used for training, where the loss is interpreted as the probability of an adversarial example. In this study, we extend the idea of adversarial detector and show that a light auxiliary network can improve the adversarial robustness.

### 2.2. Information maximization principles

Following the principle of maximum information preservation in (Linsker, 1988) and the information maximization approach in (Bell & Sejnowski, 1995), Hjelm et al. (2018) demonstrated that unsupervised learning of representations is possible by maximizing mutual information between a lower layer's representation $h_{low}(x)$ and a higher layer's representation $h_{high}(x)$ for a given in-

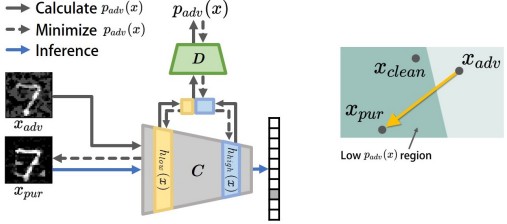

*Figure 2.* Algorithm overview and conceptual relationship of AID-Purifier. $D$ is the discriminator network.

put image $x$. Unfortunately, the precise estimation of mutual information is known to be difficult (McAllester & Stratos, 2020; Song & Ermon, 2020). A known workaround for this problem is to evaluate BCE loss or Jensen-Shannon divergence between the positive example pairs of $(h_{low}(x_i), h_{high}(x_i))$ and the negative example pairs of $(h_{low}(x_i), h_{high}(x_j))$ (Hjelm et al., 2018; Brakel & Bengio, 2017; Veličković et al., 2019; Ravanelli & Bengio, 2018), known as contrastive learning (Hadsell et al., 2006). In our AID-Purifier, we also use a BCE loss but we discriminate between $(h_{low}(x_{adv}), h_{high}(x_{adv}))$ and $(h_{low}(x_{clean}), h_{high}(x_{clean}))$ instead. This can be a natural choice for adversarial defense, because the information theoretic relationship between $h_{low}(x)$ and $h_{high}(x)$ should be different for clean examples and adversarial examples. In particular, the perturbation of features induced by $x_{adv}$ increases gradually as it passes through the network (Guo et al., 2017; Liao et al., 2018; Xie et al., 2019). The auxiliary network is denoted as $D(h_{low}(x), h_{high}(x))$.

### 2.3. AVmixup

Zhang et al. (2018) proposed mixup that is a data augmentation scheme with linearly interpolated training examples for regularizing deep networks. Mixup can be considered as a derivative of label smoothing (Szegedy et al., 2016). As a variant of the mixup, Lee et al. (2020) proposed AVmixup for performing data augmentation of adversarial examples. While AVmixup was shown to be effective for the adversarial training of the main classification network $C(x)$, we apply AVmixup to train the auxiliary discriminator network $D(h_{low}(x), h_{high}(x))$. This data augmentation with linear interpolation plays a pivotal role in training AID-Purifier. As a basic iterative procedure is applied at the inference time for purification, the discriminator needs to learn how to purify not only a strong adversarial example but also a weak adversarial example. Ideally, we would like the discriminator to learn a continuous path for purifying an adversarial

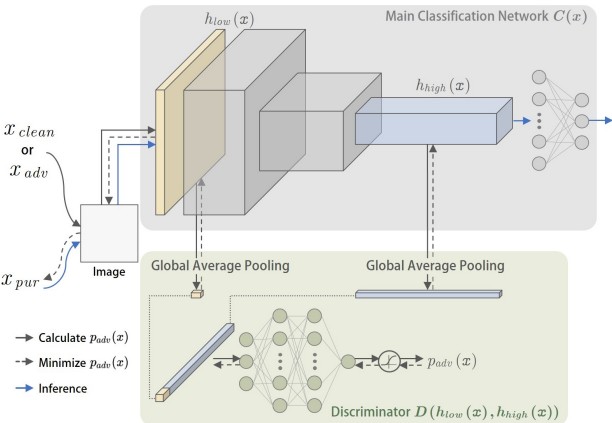

*Figure 3.* AID-Purifier. An auxiliary network $D$ is attached to the main classification network $C$.

example with an iterative procedure.

## 3. AID-Purifier

### 3.1. Discriminator: $D(h_{low}(x), h_{high}(x))$

A diagram of AID-Purifier is shown in Figure 3. The main classification network $C(x)$ can be any naturally or adversarially-trained network. The main network was frozen before attaching our auxiliary discriminator network $D(h_{low}(x), h_{high}(x))$. Following the information maximization principles, a lower layer representation $h_{low}(x)$ and a higher layer representation $h_{high}(x)$ are passed from classifier $C$ to discriminator $D$. In contrast to the work in (Hjelm et al., 2018), we apply global average pooling as the first operation in the discriminator. Despite the loss of spatial resolution in each feature map, global average pooling is helpful for making $D$ computationally light for two reasons. First, the size of the representation is significantly reduced by averaging the spatial dimensions where purification can still enforce spatial variations over the channels. Second, as the resulting representations are invariant to spatial translations (Lin et al., 2013), we can simply use a

fully connected network of a small size as the discriminator. Fully connected layers follow the global average pooling. The discriminator is trained with a standard BCE loss over adversarial and clean examples, and the loss function is as $\mathcal{L}_D = -t \log(D(h_{low}(x), h_{high}(x))) - (1-t) \log(1 - D(h_{low}(x), h_{high}(x)))$, where $x$ is the input example and $t$ is the corresponding binary label (adversarial or clean). See Appendix A for the architecture details.

### 3.2. Training: AVmixup

To train the discriminator, we apply AVmixup (Lee et al., 2020) such that the discriminator learns how to purify any strength of adversarial example. As explained in Section 2.3, this is an essential requirement for the iterative purification procedure to work well. The details of the AVmixup training are shown in Algorithm 1. We use only PGD to generate adversarial examples because PGD is the worst case attack for most scenarios.

### 3.3. Inference: iterative purification

For inference, the auxiliary discriminator network $D$ is used to purify $x$ into $x_{pur}$. The purification is applied to any $x$ including both clean and adversarial examples. As in the PGD attack, a basic iterative procedure is applied, and the purification is summarized in Algorithm 2. Specifically, an iterative gradient sign method is applied with the goal of reducing the probability of an adversarial attack, $p_{adv}(x)$. We constrain the algorithm to keep the purified image $x_{pur}$ within the $\varepsilon$-ball of $x$, because an $x_{pur}$ far from $x$ might alter the class output of the main classification network $C(x)$.

## 4. Experiments

It is certainly possible to use a purifier as a stand-alone defense, but some of the purifiers can be also used as an add-on defense for boosting the performance of another adversarial defense. In this section, we investigate the performance of AID-Purifier as a stand-alone defense and as an add-on defense. As explained in Section 1, MagNet is vulnerable to auxiliary-aware attack and SOAP cannot be used as an add-on. Thus, we only focus on Defense-GAN and PixelDefend. For the baseline adversarial training models of add-on experiments, we use Madry (Madry et al., 2017), Zhang (Zhang et al., 2019), and Lee (Lee et al., 2020) as the

---

**Algorithm 1** AVmixup training of discriminator $D$.

**Input:** Dataset $S$, input example $x$, main classifier $C$ with weights $\theta_C$, main classification label $y$, scaling factor $\gamma$, number of epochs $E$, learning rate $lr$, batch size $B$, $t_{clean} \triangleq 0, t_{adv} \triangleq 1$
**Output:** Discriminator $D$ with weights $\theta_D$
Freeze $\theta_C$, initialize $\theta_D$ of network $D$
**for** $e=1, \ldots, E$ **do**
  **for** mini-batch $\{x, y\} \sim S$ **do**
    $\delta \leftarrow PGD(x, y; \theta_C)$
    **AVmixup:**
    $x_{AV} \leftarrow x + \gamma \cdot \delta$
    $u \sim Uniform(0, 1)$
    $\hat{x} \leftarrow u \cdot x + (1-u) \cdot x_{AV}$
    $\hat{t} \leftarrow u \cdot t_{clean} + (1-u) \cdot t_{adv}$
    **Model update:**
    $\theta_D \leftarrow \theta_D - lr \cdot \frac{1}{B} \Sigma_{i=1}^{B} \nabla_{\theta_D} \mathcal{L}_D(D(h_{low}(\hat{x}), h_{high}(\hat{x})), \hat{t})$
  **end for**
**end for**

---

**Algorithm 2** Purification at inference time

1: **Input:** Main classifier $C$, discriminator $D$, input example $x$, number of iteration $N$, step size $\alpha$, epsilon $\varepsilon$
2: **Output:** Purified example $x_{pur}$
3: $x_{pur} \leftarrow x$
4: **for** n=1, \ldots, N **do**
5:   $x_{pur} \leftarrow x_{pur} - \alpha \cdot sign(\nabla_{x_{pur}} D(h_{low}(x_{pur}), h_{high}(x_{pur})))$
6:   $x_{pur} \leftarrow clip(x_{pur}, x - \varepsilon, x + \varepsilon)$
7:   $x_{pur} \leftarrow clip(x_{pur}, 0, 1)$
8: **end for**

*Table 1.* Robust accuracy: Stand-alone and add-on performances are shown for the *worst* white-box attack for SVHN.

| | Stand-alone | Add-on | | |
|---|---|---|---|---|
| | Natural training | Madry | Zhang | Lee |
| No purification | 0.01 | 22.63 | 36.72 | 46.17 |
| Defense-GAN | 41.89 | 28.42 | 38.60 | 43.42 |
| PixelDefend | 23.34 | 52.83 | 55.42 | 64.14 |
| **AID-Purifier (Ours)** | 29.10 | 49.85 | 44.76 | 62.70 |
| PixelDefend + **AID-Purifier (Ours)** | **42.67** | **64.35** | **56.68** | **65.61** |

*Table 2.* Computational load: Purification time (i.e., inference time) and training time of the adversarial purifiers. Purification time was measured with batch size one. The reported values were measured with a single RTX2080ti, except for the training time of PixelDefend's TinyImageNet that was measured with four RTX2080ti's due to the memory requirement.

| | SVHN | | CIFAR-10 | | CIFAR-100 | | TinyImageNet | |
|---|---|---|---|---|---|---|---|---|
| | Purification time (sec/img) | Training time (min) | Purification time (sec/img) | Training time (min) | Purification time (sec/img) | Training time (min) | Purification time (sec/img) | Training time (min) |
| Defense-GAN | 0.14 | 205 | 0.13 | 197 | 0.14 | 198 | 0.31 | 1385 |
| PixelDefend | 41.97 | 1185 | 40.54 | 1056 | 40.96 | 1056 | 166.31 | *5131* |
| **AID-Purifier (Ours)** | 0.59 | 23 | 0.59 | 15 | 0.59 | 16 | 0.60 | 147 |

*Table 3.* Robust accuracy: Experiment results of PixelDefend and AID-Purifier are shown for the *worst* white-box attack.

| Method | SVHN | CIFAR-10 | CIFAR-100 | TinyImageNet |
|---|---|---|---|---|
| Natural training | 0.01 | 0.00 | 0.02 | 0.00 |
| Madry | 22.63 | 51.64 | 25.42 | 20.79 |
| Zhang | 36.72 | 55.32 | 28.51 | 20.96 |
| Lee | 46.17 | 46.44 | 27.35 | 26.91 |
| Natural training + PixelDefend | 23.34 | 29.41 | 20.78 | 0.66 |
| Madry + PixelDefend | 52.83 | 54.75 | 27.34 | 21.81 |
| Zhang + PixelDefend | 55.42 | 56.68 | 30.61 | 24.21 |
| Lee + PixelDefend | 64.14 | 51.83 | 30.82 | 29.81 |
| Natural training + **AID-Purifier (Ours)** | 29.10 | 1.35 | 1.72 | 0.61 |
| Madry + **AID-Purifier (Ours)** | 49.85 | 52.65 | 27.71 | 21.23 |
| Zhang + **AID-Purifier (Ours)** | 44.76 | 56.05 | 30.58 | 24.33 |
| Lee + **AID-Purifier (Ours)** | 62.70 | 49.13 | 29.35 | **30.97** |
| Natural training + PixelDefend + **AID-Purifier (Ours)** | 42.67 | 35.82 | 23.89 | 2.62 |
| Madry + PixelDefend + **AID-Purifier (Ours)** | 64.35 | 55.07 | 28.70 | 21.81 |
| Zhang + PixelDefend + **AID-Purifier (Ours)** | 56.68 | **57.22** | 30.99 | 24.54 |
| Lee + PixelDefend + **AID-Purifier (Ours)** | **65.61** | 53.33 | **32.43** | 30.56 |

most representative set of defense models. Implementation details are described in Appendix A.

**Robust accuracy:** The robust accuracy results for SVHN are shown in Table 1. Defense-GAN performs well as a stand-alone, but its performance as an add-on is inferior to the other purifiers. In fact, as an add-on, it usually undermines the baseline performance for complex datasets as shown in Appendix B. For this reason, Defense-GAN is not investigated any further for the add-on performance. When both of PixelDefend and AID-Purifier are utilized together, however, they achieve $42.67\%$ of robust accuracy that is better than Madry or Zhang as a stand-alone. As an add-on, the best performance is also achieved when both are utilized together. This synergy is due to the diversity between the two purifiers. While PixelDefend attempts to purify an example by pushing it to a high $p_{data}(x)$ region, AID-Purifier is more adventurous because it is willing to utilize out-of-distribution regions as well.

**Computational load:** We have measured the purification time and training time of the purifiers, and the results are shown in Table 2. For the purification time, both Defense-GAN and AID-Purifier perform well but PixelDefend is up to 277 times slower than AID-Purifier due to the pixel-wise operation of PixelDefend. For the training time, AID-Purifier is definitely faster than the other two purifiers.

**Boosting performance as an add-on:** As a deep dive, we provide add-on experiment results for PixelDefend and AID-Purifier in Table 3. Exhaustive results are provided in Appendix C. The most important finding is that each of the two adversarial purifiers provides a positive enhancement for almost any individual combination of dataset and attack method. To investigate if AID-Purifier can create a positive synergy with purifiers other than PixelDefend, we have

carried out an extra experiment. For NRP, that is a type of adversarial purifier and trains a conditional GAN to learn an optimal input processing function, the experimental results are shown in Table 9 of Appendix D and they are similar to PixelDefend's results. For the case of Defense-GAN, we have tried using AID-Purifier together with Defense-GAN, and we have found that the loss by Defense-GAN can be mitigated by AID-Purifier as shown in Appendix G.

**Additional experiments:** We have performed auxiliary-aware attack, a complete white-box attack where both of the main classification network and the auxiliary network are known to the attacker (Shi et al., 2021), as a strong adaptive attack (Tramer et al., 2020) to show that the purification add-on is robust to the attack as shown in Figure 4 in Appendix E. Furthermore, we have investigated the influence of attack hyperparameters, attack epsilon, and the number of attack iterations for PGD attack as shown in Appendix J. We additionally investigate the sensitivities of the attack method, the defense epsilon, and the number of iterations for training discriminator. The results can be found in Appendix H and I. Moreover, we show that the black-box attack is not effective to AID-Purifier as shown in Table 10 of Appendix F. To verify that the key features of AID-Purifier are effective, we have performed ablation tests as shown in Table 15 of Appendix K. All of the three key features are helpful for enhancing the performance of AID-Purifier.

## 5. Conclusion

In this study, we have proposed AID-Purifier, a light auxiliary network for purifying adversarial examples. To the best of our knowledge, AID-Purifier is the first successful purification method that is based on a simple discriminator. It has a quite different characteristics from the previously known purifiers in terms of the purification objective, where a purified image $x_{pur}$ is allowed to lie in an out-of-distribution region. It can consistently boost the performance of adversarially-trained networks, and it can create synergies with other adversarial purifiers such as PixelDefend and NRP. Whether adversarially-trained networks should be always used with one or more adversarial purifiers remains as an open question.

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

# Supplementary materials for the paper "AID-Purifier: A Light Auxiliary Network for Boosting Adversarial Defense"

## A. Experimental details

In this appendix, we provide the details of the experiments conducted in our study. We perform the experiments over four datasets - SVHN (Netzer et al., 2011), CIFAR-10 (Krizhevsky & Hinton, 2009), CIFAR-100 (Krizhevsky & Hinton, 2009), and TinyImageNet (Le & Yang, 2015). As in other studies (Madry et al., 2017; Zhang et al., 2019), we use a 10-widen Wide-ResNet-34 (Zagoruyko & Komodakis, 2016) as the main classification network $C(x)$. For the white-box adversarial attack, we consider PGD (Madry et al., 2017), C&W (Carlini & Wagner, 2017), DeepFool (DF) (Moosavi-Dezfooli et al., 2016), and MIM (Dong et al., 2018).

### A.1. Other adversarial purifiers

The purifiers used in this paper are:

- Defense-GAN (Samangouei et al. (2018), Apache License) : `https://github.com/kabkabm/defensegan`.

- PixelDefend (Song et al. (2018), MIT License) : `https://github.com/microsoft/PixelDefend`.

### A.2. AID-Purifier : discriminator architecture

For training discriminator, we first apply global average pooling to $h_{low}(x)$ and $h_{high}(x)$. Then, we pass the results to a fully-connected network described below in Table 4. In the case of SVHN, we use three hidden layers instead of two.

*Table 4.* Discriminator architecture

| Operation | Size | Activation | Output |
|---|---|---|---|
| $h_{low} \to$ Linear | 1024 | ReLU | |
| Linear | 1024 | ReLU | Output 1 |
| $h_{high} \to$ Linear | 1024 | ReLU | |
| Linear | 1024 | ReLU | Output 2 |
| Concat (Output 1, Output 2) | 2048 | | |
| Linear | 1024 | ReLU | |
| Linear | 512 | ReLU | |
| Linear | 1 | | |
| Sigmoid | 1 | | |

### A.3. AID-Purifier : discriminator hyperparameters

**Training hyperparameters:** We train the network using SGD with learning rate $0.01$, weight decay $2e-4$, and momentum $0.9$ for 1 epoch. We use $\gamma = 2$ for SVHN and $\gamma = 1.5$ for CIFAR-10, CIFAR-100, and TinyImageNet.

**Purification hyperparameters:** For SVHN, we use $\varepsilon_{pur} = 12/255, \alpha = 3/255, N = 10$. For CIFAR-10, we use $\varepsilon_{pur} = 8/255, \alpha = 2/255, N = 10$. For CIFAR-100, we use $\varepsilon_{pur} = 16/255, \alpha = 2/255, N = 20$. For TinyImageNet, we use $\varepsilon_{pur} = 8/255, \alpha = 2/255, N = 20$.

### A.4. Attack hyperparameters

All attacks are evaluated under the $l_2$ metric for C&W and the $l_\infty$ metric for the others. For SVHN, we use the perturbation size 12/255 and the step size 2/255. For CIFAR-10, CIFAR-100, and TinyImageNet, we use the perturbation size 8/255 and the step size 1/255. We use Foolbox (Rauber et al., 2017), a third-party toolbox for evaluating adversarial robustness. All other parameters are set by Foolbox to be its default values.

## B. Comparison with other purifiers on various datasets

We repeated the same experiment of Table 1 for CIFAR-10 (Table 5), CIFAR-100 (Table 6), and TinyImageNet (Table 7).

*Table 5.* Robust accuracy: stand-alone and add-on performances of adversarial purifiers are shown for the *worst* white-box attack. CIFAR-10 dataset is evaluated below.

|  | Stand-alone | Add-on | | |
|---|---|---|---|---|
|  | Natural training | Madry et al. (2017) | Zhang et al. (2019) | Lee et al. (2020) |
| No purification | 0.00 | 51.64 | 55.32 | 46.44 |
| Defense-GAN | 11.68 | 18.29 | 17.99 | 17.63 |
| PixelDefend | 29.41 | 54.75 | 56.68 | 51.83 |
| **AID-Purifier** | 1.35 | 52.65 | 56.05 | 49.13 |
| PixelDefend + **AID-Purifier(Ours)** | **35.82** | **55.07** | **57.22** | **53.33** |

*Table 6.* Robust accuracy: stand-alone and add-on performances of adversarial purifiers are shown for the *worst* white-box attack. CIFAR-100 dataset is evaluated below.

|  | Stand-alone | Add-on | | |
|---|---|---|---|---|
|  | Natural training | Madry et al. (2017) | Zhang et al. (2019) | Lee et al. (2020) |
| No purification | 0.02 | 25.42 | 28.51 | 27.35 |
| Defense-GAN | 1.16 | 3.36 | 3.78 | 3.67 |
| PixelDefend | 20.78 | 27.34 | 30.61 | 30.82 |
| **AID-Purifier** | 1.72 | 27.53 | 30.58 | 29.35 |
| PixelDefend + **AID-Purifier(Ours)** | **23.89** | **28.70** | **30.99** | **32.43** |

*Table 7.* Robust accuracy: stand-alone and add-on performances of adversarial purifiers are shown for the *worst* white-box attack. TinyImageNet dataset is evaluated below.

|  | Stand-alone | Add-on | | |
|---|---|---|---|---|
|  | Natual training | Madry et al. (2017) | Zhang et al. (2019) | Lee et al. (2020) |
| No purification | 0.00 | 20.79 | 20.96 | 26.91 |
| Defense-GAN | **2.76** | 4.95 | 4.79 | 4.74 |
| PixelDefend | 0.66 | 21.81 | 24.21 | 29.81 |
| **AID-Purifier** | 0.61 | 21.23 | 24.33 | **30.97** |
| PixelDefend + **AID-Purifier(Ours)** | 2.62 | **21.81** | **24.54** | 30.56 |

## C. Boosting performance as an add-on

We provide exhaustive add-on experiment results for PixelDefend and AID-Purifier in Table 8.

## D. Boosting performance as an add-on to NRP

We provide exhaustive add-on experiment results for NRP (Neural Representation Purifier (Naseer et al., 2020)) and AID-Purifier in Table 9. For the worst performance column, which can be interpreted as the overall conclusion for each dataset, AID-Purifier boosts the robustness in most scenarios.

*Table 8.* Robust accuracy: Exhaustive experiment results for PixelDefend and AID-Purifier are shown for SVHN, CIFAR-10, CIFAR-100, and TinyImageNet datasets. As an add-on, each of PixelDefend and AID-Purifier provides a positive improvement for almost any individual combination of dataset and attack method. By inspecting the worst performance column of each dataset, it can be observed that PixelDefend+AID-Purifier achieves the best performance for three datasets and AID-Purifier achieves the best performance for TinyImageNet, which is the most complex dataset in our experiments.

| Method | SVHN | | | | | | CIFAR-10 | | | | | |
|---|---|---|---|---|---|---|---|---|---|---|---|---|
| | Clean | PGD | C&W | DF | MIM | Worst | Clean | PGD | C&W | DF | MIM | Worst |
| Natural training | 96.19 | 0.01 | 44.98 | 0.57 | 0.04 | 0.01 | 95.44 | 0.00 | 2.98 | 0.01 | 0.00 | 0.00 |
| Madry (Madry et al., 2017) | 67.39 | 38.17 | 59.08 | 28.34 | 22.63 | 22.63 | 88.72 | 51.64 | 84.75 | 54.81 | 52.45 | 51.64 |
| Zhang (Zhang et al., 2019) | 94.98 | 36.72 | 93.61 | 62.15 | 40.46 | 36.72 | 84.49 | 55.32 | 80.73 | 57.68 | 56.14 | 55.32 |
| Lee (Lee et al., 2020) | 97.29 | 55.64 | 94.00 | 52.45 | 46.17 | 46.17 | 90.46 | 46.44 | 86.41 | 54.14 | 49.32 | 46.44 |
| Natural training + PixelDefend | 88.46 | 37.70 | 83.68 | 82.89 | 23.34 | 23.34 | 85.45 | 40.41 | 82.13 | 81.97 | 29.41 | 29.41 |
| Madry (Madry et al., 2017) + PixelDefend | 74.56 | 52.83 | 72.66 | 74.29 | 55.77 | 52.83 | 87.31 | 54.75 | 85.63 | 72.71 | 54.88 | 54.75 |
| Zhang (Zhang et al., 2019) + PixelDefend | 93.03 | 55.42 | 91.73 | 90.30 | 58.14 | 55.42 | 83.41 | 56.68 | 81.61 | 68.39 | 56.89 | 56.68 |
| Lee (Lee et al., 2020) + PixelDefend | 94.03 | 64.14 | 92.71 | 89.66 | 73.92 | 64.14 | 89.01 | 51.83 | 87.29 | 69.26 | 53.29 | 51.83 |
| Natural training + **AID-Purifier (Ours)** | 78.33 | 37.25 | 67.62 | 67.83 | 29.10 | 29.10 | 87.84 | 2.15 | 78.36 | 79.55 | 1.35 | 1.35 |
| Madry (Madry et al., 2017) + **AID-Purifier (Ours)** | 89.20 | 49.85 | 88.98 | 87.63 | 52.98 | 49.85 | 88.28 | 52.65 | 86.87 | 72.00 | 53.08 | 52.65 |
| Zhang (Zhang et al., 2019) + **AID-Purifier (Ours)** | 93.04 | 45.73 | 91.78 | 82.77 | 44.76 | 44.76 | 84.59 | 56.05 | 83.07 | 69.19 | 56.57 | 56.05 |
| Lee (Lee et al., 2020) + **AID-Purifier (Ours)** | 95.28 | 62.70 | 94.00 | 88.91 | 70.41 | 62.70 | 89.59 | 49.13 | 88.04 | 67.29 | 51.24 | 49.13 |
| Natural training + PixelDefend + **AID-Purifier (Ours)** | 71.32 | 49.76 | 67.44 | 67.75 | 42.67 | 42.67 | 76.27 | 41.81 | 73.25 | 72.97 | 35.82 | 35.82 |
| Madry et al. (2017) + PixelDefend + **AID-Purifier (Ours)** | 88.71 | 64.35 | 88.39 | 87.34 | 72.27 | 64.35 | 86.66 | 55.07 | 85.28 | 72.07 | 55.10 | 55.07 |
| Zhang et al. (2019) + PixelDefend + **AID-Purifier (Ours)** | 90.28 | 56.68 | 87.51 | 84.35 | 58.80 | 56.68 | 83.50 | 57.22 | 82.06 | 69.75 | 57.67 | 57.22 |
| Lee et al. (2020) + PixelDefend + **AID-Purifier (Ours)** | 93.04 | 65.61 | 91.44 | 89.07 | 74.23 | 65.61 | 87.85 | 53.33 | 86.34 | 69.67 | 54.32 | 53.33 |

| Method | CIFAR-100 | | | | | | TinyImageNet | | | | | |
|---|---|---|---|---|---|---|---|---|---|---|---|---|
| | Clean | PGD | C&W | DF | MIM | Worst | Clean | PGD | C&W | DF | MIM | Worst |
| Natural training | 78.17 | 0.02 | 3.23 | 0.04 | 0.05 | 0.02 | 64.98 | 0.02 | 20.91 | 0.00 | 0.31 | 0.00 |
| Madry (Madry et al., 2017) | 64.69 | 25.42 | 57.71 | 26.77 | 26.21 | 25.42 | 58.44 | 21.05 | 52.12 | 20.79 | 21.64 | 20.79 |
| Zhang (Zhang et al., 2019) | 56.90 | 29.87 | 51.13 | 28.51 | 30.29 | 28.51 | 50.28 | 24.11 | 44.97 | 20.96 | 24.32 | 20.96 |
| Lee (Lee et al., 2020) | 74.56 | 27.35 | 64.33 | 33.96 | 30.13 | 27.35 | 65.09 | 26.91 | 57.27 | 26.52 | 27.96 | 26.91 |
| Natural training + PixelDefend | 61.19 | 29.95 | 58.23 | 58.73 | 20.78 | 20.78 | 56.23 | 0.68 | 45.35 | 51.33 | 0.66 | 0.66 |
| Madry (Madry et al., 2017) + PixelDefend | 62.90 | 27.34 | 59.64 | 46.81 | 27.70 | 27.34 | 57.65 | 21.81 | 54.92 | 41.01 | 22.29 | 21.81 |
| Zhang (Zhang et al., 2019) + PixelDefend | 55.43 | 30.61 | 52.54 | 42.35 | 30.85 | 30.61 | 49.53 | 24.21 | 47.50 | 35.59 | 24.36 | 24.21 |
| Lee (Lee et al., 2020) + PixelDefend | 69.87 | 30.82 | 67.75 | 57.31 | 32.14 | 30.82 | 58.16 | 29.81 | 57.11 | 49.44 | 30.13 | 29.81 |
| Natural training + **AID-Purifier (Ours)** | 60.50 | 2.81 | 53.45 | 55.62 | 1.72 | 1.72 | 55.89 | 0.81 | 45.24 | 51.11 | 0.61 | 0.61 |
| Madry (Madry et al., 2017) + **AID-Purifier (Ours)** | 61.25 | 27.71 | 59.27 | 46.50 | 27.74 | 27.71 | 58.58 | 21.23 | 55.38 | 41.53 | 21.84 | 21.23 |
| Zhang (Zhang et al., 2019) + **AID-Purifier (Ours)** | 54.56 | 30.58 | 53.15 | 43.35 | 30.68 | 30.58 | 50.23 | 24.33 | 47.49 | 36.58 | 24.43 | 24.33 |
| Lee (Lee et al., 2020) + **AID-Purifier (Ours)** | 71.55 | 29.35 | 68.72 | 62.62 | 31.17 | 29.35 | 63.52 | 31.03 | 59.32 | 55.87 | 30.97 | 30.97 |
| Natural training + PixelDefend + **AID-Purifier (Ours)** | 50.88 | 27.74 | 48.79 | 49.01 | 23.89 | 23.89 | 49.26 | 4.34 | 43.16 | 45.76 | 2.62 | 2.62 |
| Madry (Madry et al., 2017) + PixelDefend + **AID-Purifier (Ours)** | 58.95 | 28.70 | 57.13 | 45.26 | 28.75 | 28.70 | 57.92 | 21.81 | 55.23 | 41.88 | 22.19 | 21.81 |
| Zhang (Zhang et al., 2019) + PixelDefend + **AID-Purifier (Ours)** | 52.93 | 30.99 | 51.62 | 42.48 | 31.18 | 30.99 | 49.76 | 24.54 | 47.63 | 36.92 | 24.66 | 24.54 |
| Lee (Lee et al., 2020) + PixelDefend + **AID-Purifier (Ours)** | 67.89 | 32.43 | 66.06 | 58.98 | 33.01 | 32.43 | 59.10 | 30.56 | 57.96 | 51.10 | 30.63 | 30.56 |

# E. Auxiliary-aware attack

Following the approach in (Shi et al., 2021), we generate auxiliary-aware attack as

$$x_{adv} \leftarrow \underset{x_{adv} \in \mathcal{N}(x)}{argmax} \; \mathcal{L}_C - \lambda \cdot \mathcal{L}_D, \tag{1}$$

where $\lambda$ is a trade-off parameter between the main cross entropy loss $\mathcal{L}_C$ and the auxiliary (discriminator) loss $\mathcal{L}_D$. The results are shown in Figure 4. Purification provides positive improvement for all the evaluated cases.

# F. Black-box attack

We generate black-box adversarial examples using a pre-trained VGG19 network (Simonyan & Zisserman, 2014). PGD, C&W, DF, and MIM attacks are used to evaluate black-box robustness on SVHN. We report only the worst black-box accuracy in Table 10. For the case of Mardy, a large improvement is achieved.

# G. Boosting performance of AID-Purifier as an add-on to the Defense-GAN

The robust accuracy results are shown in Table 11. Defense-GAN is actually harmful, but AID-Purifier can recover part of the performance loss created by Defense-GAN.

*Table 9.* Robust accuracy: exhaustive experiment results for NRP and AID-Purifier are shown for SVHN, CIFAR-10, CIFAR-100, and TinyImageNet datasets. As an add-on, each of NRP and AID-Purifier provides a positive improvement for almost any individual combination of dataset and attack method. By inspecting the worst performance column of each dataset, it can be observed that NRP+AID-Purifier achieves the best performance for two datasets, CIFAR-10 and CIFAR-100, and AID-Purifier achieves the best performance for two datasets, SVHN and TinyImageNet. Results of stand-alone and add-on using AID-Purifier only are duplicated from Table 3.

| Method | SVHN | | | | | | CIFAR-10 | | | | | |
|---|---|---|---|---|---|---|---|---|---|---|---|---|
| | Clean | PGD | C&W | DF | MIM | Worst | Clean | PGD | C&W | DF | MIM | Worst |
| Natural training | 96.19 | 0.01 | 44.98 | 0.57 | 0.04 | 0.01 | 95.44 | 0.00 | 2.98 | 0.01 | 0.00 | 0.00 |
| Madry (Madry et al., 2017) | 67.39 | 38.17 | 59.08 | 28.34 | 22.63 | 22.63 | 88.72 | 51.64 | 84.75 | 54.81 | 52.45 | 51.64 |
| Zhang (Zhang et al., 2019) | 94.98 | 36.72 | 93.61 | 62.15 | 40.46 | 36.72 | 84.49 | 55.32 | 80.73 | 57.68 | 56.14 | 55.32 |
| Lee (Lee et al., 2020) | 97.29 | 55.64 | 94.00 | 52.45 | 46.17 | 46.17 | 90.46 | 46.44 | 86.41 | 54.14 | 49.32 | 46.44 |
| Natural training + NRP | 88.42 | 33.52 | 80.44 | 77.81 | 19.70 | 19.70 | 70.95 | 41.02 | 62.82 | 62.79 | 6.60 | 6.60 |
| Madry (Madry et al., 2017) + NRP | 82.29 | 49.54 | 81.05 | 81.44 | 51.83 | 49.54 | 86.08 | 44.02 | 84.06 | 70.69 | 54.96 | 44.02 |
| Zhang (Zhang et al., 2019) + NRP | 93.00 | 52.71 | 91.68 | 84.94 | 50.63 | 50.63 | 81.90 | 55.83 | 79.69 | 66.04 | 56.15 | 55.83 |
| Lee (Lee et al., 2020) + NRP | 94.23 | 62.68 | 92.71 | 83.51 | 65.48 | 62.68 | 87.26 | 52.98 | 85.60 | 69.02 | 54.46 | 52.98 |
| Natural training + **AID-Purifier (Ours)** | 78.33 | 37.25 | 67.62 | 67.83 | 29.10 | 29.10 | 87.84 | 2.15 | 78.36 | 79.55 | 1.35 | 1.35 |
| Madry (Madry et al., 2017) + **AID-Purifier (Ours)** | 89.20 | 49.85 | 88.98 | 87.63 | 52.98 | 49.85 | 88.28 | 52.65 | 86.87 | 72.00 | 53.08 | 52.65 |
| Zhang (Zhang et al., 2019) + **AID-Purifier (Ours)** | 93.04 | 45.73 | 91.78 | 82.77 | 44.76 | 44.76 | 84.59 | 56.05 | 83.07 | 69.19 | 56.57 | 56.05 |
| Lee (Lee et al., 2020) + **AID-Purifier (Ours)** | 95.28 | **62.70** | 94.00 | **88.91** | **70.41** | **62.70** | 89.59 | 49.13 | **88.04** | 67.29 | 51.24 | 49.13 |
| Natural training + NRP + **AID-Purifier (Ours)** | 66.89 | 42.09 | 62.72 | 62.59 | 36.02 | 36.02 | 66.38 | 40.58 | 61.04 | 60.86 | 20.46 | 20.46 |
| Madry et al. (2017) + NRP + **AID-Purifier (Ours)** | 85.33 | 53.59 | 84.62 | 83.16 | 56.49 | 53.59 | 86.14 | 46.27 | 84.53 | 70.69 | 55.31 | 46.27 |
| Zhang et al. (2019) + NRP + **AID-Purifier (Ours)** | 87.02 | 53.39 | 83.65 | 73.59 | 50.27 | 50.27 | 82.22 | **56.63** | 81.03 | 67.26 | **56.71** | **56.63** |
| Lee et al. (2020) + NRP + **AID-Purifier (Ours)** | 92.05 | 61.97 | 90.35 | 81.34 | 65.22 | 61.97 | 86.25 | 53.92 | 84.40 | 68.57 | 55.06 | 53.92 |

| Method | CIFAR-100 | | | | | | TinyImageNet | | | | | |
|---|---|---|---|---|---|---|---|---|---|---|---|---|
| | Clean | PGD | C&W | DF | MIM | Worst | Clean | PGD | C&W | DF | MIM | Worst |
| Natural training | 78.17 | 0.02 | 3.23 | 0.04 | 0.05 | 0.02 | 64.98 | 0.02 | 20.91 | 0.00 | 0.31 | 0.00 |
| Madry (Madry et al., 2017) | 64.69 | 25.42 | 57.71 | 26.77 | 26.21 | 25.42 | 58.44 | 21.05 | 52.12 | 20.79 | 21.64 | 20.79 |
| Zhang (Zhang et al., 2019) | 56.90 | 29.87 | 51.13 | 28.51 | 30.29 | 28.51 | 50.28 | 24.11 | 44.97 | 20.96 | 24.32 | 20.96 |
| Lee (Lee et al., 2020) | 74.56 | 27.35 | 64.33 | 33.96 | 30.13 | 27.35 | 65.09 | 26.91 | 57.27 | 26.52 | 27.96 | 26.91 |
| Natural training + NRP | 40.25 | 6.23 | 35.93 | 36.45 | 5.85 | 5.85 | 49.28 | 9.13 | 43.41 | 45.71 | 7.22 | 7.22 |
| Madry (Madry et al., 2017) + NRP | 61.49 | 27.43 | 58.78 | 45.7 | 27.75 | 27.43 | 55.69 | 22.78 | 53.78 | 40.2 | 23.24 | 22.78 |
| Zhang (Zhang et al., 2019) + NRP | 54.74 | 30.25 | 52.58 | 42.44 | 30.55 | 30.25 | 47.69 | 23.98 | 45.80 | 33.30 | 24.00 | 23.98 |
| Lee (Lee et al., 2020) + NRP | 65.14 | 32.76 | 63.41 | 54.25 | **33.68** | 32.76 | 55.38 | 29.75 | 54.52 | 47.25 | 30.39 | 29.75 |
| Natural training + **AID-Purifier (Ours)** | 60.50 | 2.81 | 53.45 | 55.62 | 1.72 | 1.72 | 55.89 | 0.81 | 45.24 | 51.11 | 0.61 | 0.61 |
| Madry (Madry et al., 2017) + **AID-Purifier (Ours)** | 61.25 | 27.71 | 59.27 | 46.50 | 27.74 | 27.71 | 58.58 | 21.23 | 55.38 | 41.53 | 21.84 | 21.23 |
| Zhang (Zhang et al., 2019) + **AID-Purifier (Ours)** | 54.56 | 30.58 | 53.15 | 43.35 | 30.68 | 30.58 | 50.23 | 24.33 | 47.49 | 36.58 | 24.43 | 24.33 |
| Lee (Lee et al., 2020) + **AID-Purifier (Ours)** | 71.55 | 29.35 | **68.72** | 62.62 | 31.17 | 29.35 | 63.52 | **31.03** | **59.32** | 55.87 | 30.97 | **30.97** |
| Natural training + NRP + **AID-Purifier (Ours)** | 37.59 | 13.53 | 35.15 | 35.25 | 12.08 | 12.08 | 40.87 | 15.65 | 37.45 | 38.34 | 12.96 | 12.96 |
| Madry (Madry et al., 2017) + NRP + **AID-Purifier (Ours)** | 59.17 | 28.31 | 57.04 | 45.97 | 28.68 | 28.31 | 55.86 | 22.89 | 53.97 | 40.78 | 23.27 | 22.89 |
| Zhang (Zhang et al., 2019) + NRP + **AID-Purifier (Ours)** | 47.76 | 24.17 | 45.95 | 33.63 | 23.95 | 23.95 | 47.76 | 24.17 | 45.95 | 33.63 | 23.95 | 23.95 |
| Lee (Lee et al., 2020) + NRP + **AID-Purifier (Ours)** | 64.73 | **32.86** | 63.28 | 54.72 | 33.49 | **32.86** | 58.07 | 30.70 | 56.73 | 50.01 | 30.94 | 30.70 |

## H. Various attack methods for training discriminator $D$

When training the discriminator, the attack method for generating adversarial examples need to be decided. The sensitivity study results are shown in Table 12 for SVHN dataset.

## I. Defense epsilon and defense iteration

The robust accuracy results for SVHN are shown in Figure 5 and Table 13.

## J. Attack epsilon and attack iteration

The robust accuracy results for SVHN are shown in Table 14.

## K. Ablation study

To verify that the key features of AID-Purifier are effective, we have performed ablation tests. The results are shown in Table 15. All of the three key features are helpful for enhancing the performance of AID-Purifier, where the choice of training target and the choice of data augmentation scheme are crucial for AID-Purifier's performance. Additionally, the

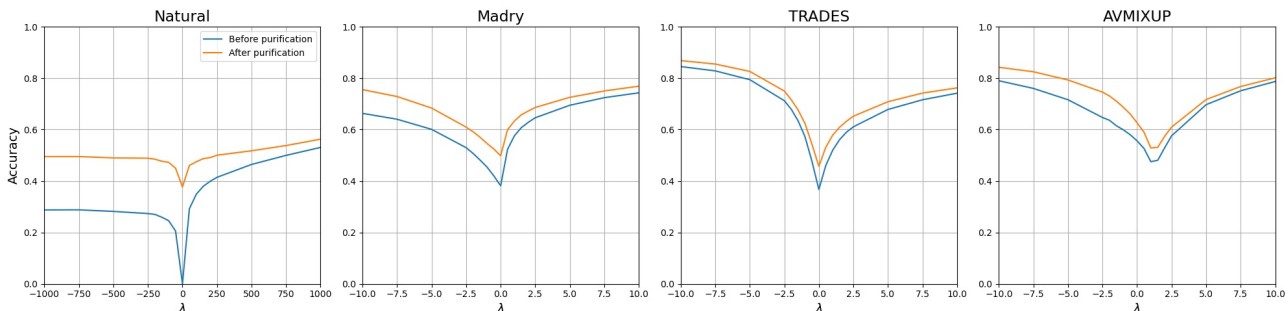

*Figure 4.* Robust accuracy: robustness before (blue) and after (orange) purification are shown against auxiliary-aware PGD attacks.

*Table 10.* Robust accuracy: stand-alone and add-on performances of adversarial purifiers are shown for the worst black-box attack on SVHN.

|  | Worst black-box attack | |
| --- | --- | --- |
| Training method | No purification | **AID-Purifier (Ours)** |
| Natural training | 45.26 | 47.93 |
| Madry (Madry et al., 2017) | 62.52 | 80.74 |
| Zhang (Zhang et al., 2019) | 85.14 | 80.71 |
| Lee (Lee et al., 2020) | 79.94 | 79.69 |

ablation test results of the intermediate layers connected from $C(x)$ to the discriminator $D$ are presented in Table 16.

*Table 11.* Robust accuracy. Experiment results for Defense-GAN and AID-Purifier are shown (SVHN under PGD attack; Madry is used to train the main classification network).

|  | PGD |
| --- | --- |
| No purification | 38.17 |
| Defense-GAN | 28.59 |
| Defense-GAN + AID-Purifier | 32.29 |

*Table 12.* Attack method used at the time of training and the resulting robust accuracy (SVHN under PGD attack; Madry is used to train the main classification network).

| AID-Purifier | PGD | C&W | DF | Worst |
| --- | --- | --- | --- | --- |
| PGD training (our work) | **37.25** | 67.62 | 67.83 | **37.25** |
| C&W training | 11.36 | 52.27 | 45.73 | 11.36 |
| DF training | 0.39 | **74.12** | **74.65** | 0.39 |
| PGD + CW training | 8.51 | 54.77 | 48.85 | 8.51 |
| PGD + DF training | 1.16 | 68.01 | 69.74 | 1.16 |
| CW + DF training | 19.14 | 68.73 | 64.16 | 19.14 |
| PGD + CW + DF training | 8.73 | 67.83 | 62.94 | 8.73 |

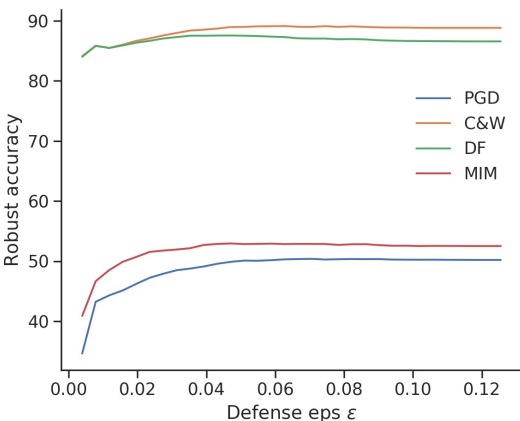

| Number of iterations | Madry+AID-Purifier |
| --- | --- |
| 20 | 50.62 |
| 10 (our work) | 49.85 |
| 5 | 49.26 |
| 4 | 48.95 |
| 2 | 46.68 |

*Figure 5.* Robust accuracy for Madry+AID-Purifier is shown with respect to the variations in the defense epsilon (SVHN under PGD attack; Madry is used to train the main classification network).

*Table 13.* Robust accuracy for Madry+AID-Purifier with respect to the variations in the number of iterations (SVHN under PGD attack; Madry is used to train the main classification network).

*Table 14.* Robust accuracy: (SVHN under PGD attack; Madry is used to train the main classification network) (a) Performance of Madry and Madry+AID-Purifier for varying the attack epsilon of PGD are shown for SVHN. (b) Performance of Madry and Madry+AID-Purifier for varying the attack iteration of PGD are shown for SVHN.

| Attack eps | Madry | Madry + AID-Purifier |
| --- | --- | --- |
| 1/255 | 57.29 | 88.20 |
| 2/255 | 49.49 | 86.92 |
| 4/255 | 42.18 | 83.82 |
| 8/255 | 34.94 | 73.14 |
| 12/255 (our work) | 38.17 | 49.85 |
| 16/255 | 22.28 | 28.62 |
| 32/255 | 1.27 | 1.59 |

| | Madry | Madry + AID-Purifier |
| --- | --- | --- |
| 40 (our work) | 38.17 | 49.85 |
| 100 | 35.64 | 48.60 |
| 200 | 34.86 | 48.34 |

(a)

(b)

*Table 15.* Ablation test results over the key features of AID-Purifier: The evaluations are over Madry, SVHN, and PGD. The baseline performance of Madry without any add-on is 38.17%. (a) Number of intermediate layers connected from $C(x)$ to the discriminator. (b) Training targets. (c) Data augmentation method for training.

| Number of $h(x)$ | Accuracy (%) |
|:---:|:---:|
| 1 | 48.93 |
| 2 | **49.85** |

(a) Number of $h(x)$

| Targets | Accuracy (%) |
|:---|:---:|
| Contrastive | 44.87 |
| Clean vs. adv. | **49.85** |

(b) Training targets

| Augmentation | Accuracy (%) |
|:---|:---:|
| None | 46.02 |
| AV | 47.73 |
| Mixup | 48.49 |
| AVmixup | **49.85** |

(c) Data augmentation

*Table 16.* Ablation test of the intermediate layers connected from $C(x)$ to the discriminator (SVHN under PGD attack; Madry is used to train the main classification network). 1st Conv denotes the output of the first convolution layer and n-th Block denotes the output of the n-th residual block, where downsampling is performed. Check symbols indicate the connected layers. The best performing combination is {10th block, 15th block}, but we have used {1st Conv, 15th block} in our main experiment.

| Number of $h(x)$ | \multicolumn{4}{Used intermediate representation} | Worst white-box attack |
|:---:|:---:|:---:|:---:|:---:|:---:|
| | 1st Conv | 5th Block | 10th Block | 15th Block | **AID-Purifier (Ours)** |
| 1 | ✓ | | | | 48.28 |
| | | ✓ | | | 48.66 |
| | | | ✓ | | 48.92 |
| | | | | ✓ | 48.93 |
| 2 | ✓ | ✓ | | | 44.43 |
| | ✓ | | ✓ | | 48.97 |
| | ✓ | | | ✓ | 49.85 |
| | | ✓ | ✓ | | 44.51 |
| | ✓ | | | ✓ | 45.77 |
| | | | ✓ | ✓ | **50.10** |