# OpenReview forum: "AID-Purifier: A Light Auxiliary Network for Boosting Adversarial Defense"
_ICML.cc/2021/Workshop/AML — ICML 2021 Workshop AML Poster_

### Official Review · Reviewer_hqFc · 2021-06-19
**The paper proposed AID-purifier, which is a light auxiliary network and can boost the robustness of adversarially-trained networks by purifying their inputs.**

**Rating:** Accept
**Confidence:** 3

**Review:**

The paper considered adversarial purification for adversarial defense and proposed a computationally light and easily attachable purifier: AID-Purifier, which is the first successful purification method that is based on a simple discriminator. AID-Purifier trains its discriminator with the BCE loss between $x_{adv}$ and $x_{clean}$ with AVmixup and inference with iterative purifification. Extensive experiments show AID-Purifier can improve the robustness of the model as well as decreasing the inference time significantly. The method in this paper is effective and the experiment is solid.

---

### Decision · Program_Chairs · 2021-06-21

**Decision:**

Accept (Poster)

**Comment:**

This paper proposed an adversarial purification method for defense. The reviewer agreed that the paper is solid and the experiments are extensive.